# High Temperature Deformation Behavior and Microstructure Evolution of Low-Density Steel Fe30Mn11Al1C Micro-Alloyed with Nb and V

**DOI:** 10.3390/ma14216555

**Published:** 2021-11-01

**Authors:** Hui Wang, Ziyuan Gao, Zhiyue Shi, Haifeng Xu, Ling Zhang, Guilin Wu, Chang Wang, Cunyu Wang, Yuqing Weng, Wenquan Cao

**Affiliations:** 1College of Materials Science and Engineering, Chongqing University, Chongqing 400044, China; wanghui2018@cqu.edu.cn (H.W.); zhangling2014@cqu.edu.cn (L.Z.); 2Special Steel Department of Central Iron and Steel Research Institute (CISRI), Beijing 100081, China; gzy940927@gmail.com (Z.G.); B20190517@xs.ustb.edu.cn (Z.S.); xuhaifeng@nercast.com (H.X.); wangchang@nercast.com (C.W.); wangcunyu@nercast.com (C.W.); wengyuqing@nercast.com (Y.W.); 3Beijing Advanced Innovation Center for Materials Genome Engineering, University of Science and Technology Beijing, Beijing 100083, China; guilinwu@ustb.edu.cn

**Keywords:** low-density steel, high temperature deformation, thermal processing map, dynamic recrystallization, continuous dynamic recrystallization

## Abstract

The thermal processing parameters is very important to the hot rolling and forging process for producing grain refinement in lightweight high-manganese and aluminum steels. In this work, the high temperature deformation behaviors of a low-density steel of Fe30Mn11Al1C alloyed with 0.1Nb and 0.1V were studied by isothermal hot compression tests at temperatures of 850–1150 °C and strain rates between 0.01 s^−1^ and 10 s^−1^. It was found that the flow stress constitutive model could be effectively established by the Arrhenius based hyperbolic sine equation with an activation energy of about 389.1 kJ/mol. The thermal processing maps were developed based on the dynamic material model at different strains. It’s shown that the safe region for high temperatures in a very broad range of both deformation temperature and deformation strain and only a small unstable high deformation region, located at low temperatures lower than 950 °C. The deformation microstructures were found to be fully recrystallized microstructure in the safe deformation region and the grain size decreases along with decreasing temperature and increasing strain rate. Whereas the deformation microstructures is composed by grain refinement-recrystallized grains and a small fraction of non-recrystallized microstructure in the unstable deformation region, indicating that the deformation behaviors controlled by continuous dynamic recrystallization. The Hall Petch relationship between microhardness and the grain size of the high temperature deformed materials indicates that high strength low-density steel could be developed by a relative low temperature deformation and high strain rate.

## 1. Introduction

Facing with the requirements of weight-lightening of metallic equipment, low-density steel has attention because of its significantly reduced density, high ductility, high toughness and possible applications in engineering [1,2,3]. It has been shown that the density reduction of austenitic low-density steel is mainly lattice expansion and the addition of light elements, such as carbon(C) and aluminum(Al) [4,5,6]. Kalashnikov et al. [7] researched the influence of Al and C content on the tensile strength and U-notch impact toughness of Fe-30Mn-*x*Al-0.95C austenite steels in 550 °C aged condition. The impact toughness reduced significantly with the Al content increasing. Acselrad et al. [8] have studied the impact toughness in solutionized and aged condition of Fe-30Mn-10Al-1C-1Si alloy. The materials in the solution treated condition show a high charpy V-notch impact energy of 200 J with a yield strength of 452 MPa at room temperature, while the impact energy reduced to almost zero in aged condition at 550 °C for 16 h. High Al content makes it difficult to obtain both high ductility and high toughness by aging process.

For structural purposes, a good combination of mechanical strength and fracture toughness is desirable. In order to obtain both high ductility and high toughness, low-density steel has to be solution treated to suppress κ-carbides, which takes place in the cooling or aging process [2,3,9,10]. At the same time, the solution treatment induces the significant grain growth and therefore decreases both yield and tensile strength according to the classical Hall Petch law [11,12]. Ren et al. [11] produced a series of grain refinement alloys by cold rolling reduction of 90% and recrystallization annealing, high yield strength of 945 MPa and 47% elongation were obtained for Fe-30Mn-11Al-1.2C steel with an average grain size of 2.2 μm, which is about 350 MPa improvement than the coarsening condition. Thus, it is very important to control the κ-carbide precipitation and austenite grain size by the chemical composition design and the selection of the thermal processing parameters for obtaining a very good combination of strength, ductility and toughness.

Different from the traditional austenitic steel, such as 304 austenitic stainless steels and twin induced plasticity (TWIP) steels, the low-density Fe-Mn-Al-C steel was designed with the considerable of light element addition, such as 6–12% Al, 20–30% Mn and 0.80–1.50% carbon [13,14,15,16,17,18,19,20,21]. The addition of Al results in the high stacking fault energy (SFE) and makes dislocation planar slipping [1,11,13,14,20,21]. Very importantly, the addition of Al also induces dynamic precipitation of κ-carbide in the face centered cubic (FCC) structure by the spinodal decomposition of austenite during cooling and aging process, which deteriorates the ductility and toughness [7,22,23,24]. In order to overcome this drawback, Nb was proposed to addition to the austenitic Fe-Mn-Al-C steel because Nb has a strong affinity with C to form a very stable NbC phase during heat treatment process [25], which suppresses the precipitation of coarse κ-carbides [26] and retards the grain growth of austenite significantly due to the existence of stable nanosized NbC during the high temperature deformation process [27]. However, the effects of Nb on high temperature deformation behaviors and microstructural evolutions are seldom reported in the austenitic low-density steel.

In this work, the thermal process behaviors of an austenitic low-density steel micro-alloyed with Nb and V were studied by various strain rates and temperatures uniaxial compression tests. The effects of deformation temperature and the deformation strain rate on the thermal flow behaviors and the microstructure evolution were discussed. The objective of this work is constructing a thermal processing map of the designed low-density steel for the optimization of thermal deformation parameters to obtain the combination of strength, ductility and toughness by the austenite grain refinement above temperature of κ′-carbide precipitate [16].

## 2. Experimental Procedures

The chemical composition of Fe30Mn11Al1C0.1Nb0.1V steel is shown in Table 1. The designed steel was melted in a vacuum induction furnace with a capacity of 50 kg and cast into an ingot with diameter of 120 mm. Then the ingot was homogenized in a box-type resistance furnace at 1180 °C for 2 h, and finally forged to bars with diameter of 15 mm (finish forging temperature 950 °C), and air-cooled to room temperature. Cylindrical specimens with a diameter of 8 mm and a height of 15 mm were machined from the forged bars. The thermal compression tests were performed on a Gleeble-3800E thermal-mechanical simulator with temperature ranging from 850 °C to 1150 °C and strain rate ranging from 0.01 s^−1^ to 10 s^−1^. The compression experiments were carried out according to the processing parameters as shown in Figure 1. The specimens were heated from room temperature to the designed deformation temperature with a heating rate of 10 °C · s^−1^ and a hold time of 5 min at the deformation temperature. The compression tests were carried out at a given temperature and a given deformation rate until the true strain reaching to 0.9. After the compression deformation, the specimens were immediately quenched into cold water to reserve the microstructure at the high temperature.

Microstructures were examined by a field emission gun scanning electron microscope (FEG-SEM) with an electron backscatter diffraction (EBSD) detector. The mapping was performed at 20 kV and a step size of 0.5 μm. The average grain size was processed by Channel 5 software for grains having minimum misorientation angles of 15° (excluding twin boundaries). Samples for the EBSD examination were electropolished in a solution of 7% perchloric acid and 93% ethanol at room temperature. The microhardness test was carried out on FM-300 microhardness tester with a load of 300 g and a dwell time of 10 s.

## 3. Results

### 3.1. High Temperature Stress-Strain Curves

The true stress-strain curves obtained by thermal compression tests under different deformation temperatures and strain rates are shown in Figure 2. It can be seen from Figure 2a that with a compression strain rate of 0.01 s^−1^, the flow stress increases quickly after yielding and then decreases gradually after the peak stress, which indicating a transition of overall flow behavior from the work hardening stage to a softening stage. In addition, the yield stress, the peak stress and the transition strain from hardening to softening are increased with decreasing deformation temperature, implying that the dislocation storage capacity and the dislocation structure stability increases with decreasing deformation temperature. The steady-state flow behaviors of the Fe30Mn11Al1C0.1Nb0.1V steel deformed at different temperatures imply that the microstructure experiences a continuous evolution during high temperature deformation process. Similar to the flow behaviors at a compression strain rate of 0.01 s^−1^, the yield stress, peak stress, and transition strain all increase with decreasing deformation temperature with strain rates of 0.1 s^−1^, 1 s^−1^ and 10 s^−1^, as revealed in Figure 2b–d. Based on the comparison of the stress-strain curves of the Fe30Mn11Al1C0.1Nb0.1V steel deformed at different strain rates, the yield stress, peak stress and the transition strain increase with increasing deformation strain rate as shown in Figure 2, manifesting that the dislocation storage capacity and the deformation microstructure stability are enhanced by increasing the deformation strain rate. Beside the overall steady-state flow behaviors as shown in Figure 2, a yield drop phenomena can be found when the compression test was carried out at strain rates of 0.1 s^−1^ and 1.0 s^−1^ as shown in Figure 2b,c, implying an unstable deformation behavior in the early deformation stage, which may be resulted from the segregation of the carbon atoms into dislocations.

### 3.2. High Temperature Deformation Microstructure

The grain boundary maps of specimens deformed at the strain rate of 1 s^−1^ under deformation temperatures of 850 °C, 950 °C, and 1050 °C are shown in Figure 3. It can be seen from Figure 3a that the deformation microstructure of the specimen deformed at 850 °C is composed by 70.5% recrystallized region and 29.5% non-recrystallized. The average grain size of deformation microstructure is only 2.3 μm. When deformed at temperature up to 950 °C, recrystallized region increases to 95.3% and the non-recrystallized region decreased to 4.7%, the average grain size of deformation microstructure is 3.3 μm. In contrast to recrystallized region, the non-recrystallized regions contain a certain level of LAGBs, which is caused by work hardening. Further increasing the deformation temperature with a strain rate of 1 s^−1^, a fully recrystallized microstructure can be seen as shown in Figure 3c, resulting in an average grain size only about 5.7 μm in the specimen deformed at 1050 °C. Thus, a uniform and grain-refined deformation microstructure can be obtained after a careful selection of high deformation parameters.

In order to examine the effect of strain rate on the microstructure at 1050 °C, the microstructures of the specimens deformed at 1050 °C with different deformation rate are shown in Figure 4. It can be seen that the high temperature deformation microstructures comprise of fully recrystallized austenite grains and the grain size is refined significantly with increasing deformation strain rates. The average grain sizes are both 3.7 μm when deformed at strain rate of 1 s^−1^ and 10 s^−1^. This means that the deformation temperature of 1050 °C is very suitable for obtaining a fully recrystallized and grain refinement microstructure.

As shown in Figure 3, microstructure with an average grain size of only 2-4 μm can be obtained in the recrystallized regions of the specimens deformed at 850 °C and 950 °C. It is thus very important to find the possibility of the fully recrystallized deformation microstructure by relative low temperature deformation. The thermal deformation microstructures at 850 °C and 950 °C examined by EBSD are given in Figure 5. It can be seen that the non-recrystallized regions contain a certain level of LAGBs and the area fraction of recrystallized region increases with increasing compression strain rate as shown in Figure 5a-d for the low-density steel of Fe30Mn11Al1C0.1Nb0.1V steel deformed at 850 °C and 950 °C with different strain rates. It is seen that the area fraction of the recrystallized region is over 89.2% and that recrystallized grain size is about 2.7 μm in the specimens deformed at 850 °C and with a strain rate of 10 s^−1^. While, the area fraction of the recrystallized region is over 95.4% and the recrystallized grain size is about 3.8 μm in the specimens deformed at 950 °C and strain rate of 10 s^−1^.

## 4. Discussion

### 4.1. Constitutive Analysis of Thermal Deformation Flow Behaviors

The stress-strain curves as shown in Figure 2 are strongly affected by the deformation microstructure and deformation strain rate; and all the stress-strain curves are featured by the steady-state transition from hardened stage to softened stage. It is well known that the Arrhenius equation (Equation (1)) is widely used to describe the relationship among strain rate, flow stress and deformation temperature, especially at high temperature [28,29], which can reflect the balance between the work hardening and dynamic softening during hot deformation process, and can also be used to characterize the difficulty of thermal deformation. Also, the effects of the temperatures and strain rate on the deformation behaviors can be represented by Zener-Hollomon (Z) parameter in an exponent-type equation (Equation (2)) [29]. For different stress levels, the relationship between Zener-Hollomon (Z), the flow stress, the strain rate ($\dot{\varepsilon }$) and the deformation temperature could be described by Equation (3), (4) and (5) as following [30,31,32],
(1)$$\dot{\varepsilon }=Af\left(\sigma \right)exp\left(-\frac{Q}{RT}\right)$$
(2)$$Z=\dot{\varepsilon }exp\left(Q/RT\right)\;$$
(3)$$Z=\dot{\varepsilon }exp\left(\frac{Q}{RT\;}\right)=f\left(\sigma \right)=A_{1}\sigma^{n1}\;$$
(4)$$Z=\dot{\varepsilon }exp\left(\frac{Q}{RT\;}\right)=f\left(\sigma \right)=A_{2}exp\left(\beta \sigma \right)\;$$
(5)$$Z=\dot{\varepsilon }exp\left(\frac{Q}{RT\;}\right)=f\left(\sigma \right)=A_{3}{\left[sinh\left(\alpha \sigma \right)\right]}^{n}\;$$
where *A*_1_, *A*_2_, *A*_3_, *β*, *α*, *n*_1_ and *n* are material constants with *α* = *β*/*n*_1_, *σ* is the flow stress, *R* is the gas constant (8.314 J mol^−1^ K^−1^), *T* is the hot deformation temperature, *Q* is the deformation activation energy.

The characteristic stress can either be the initial yield stress, the peak stress (*σ_p_*), or strain (*ε_p_*) dependent flow stress. Generally, for materials prone to dynamic recovery, the steady-state stress value is often selected as the characteristic stress value. While for materials prone to dynamic recrystallization, the peak stress value is often selected as the characteristic stress value [32,33]. In this study, the peak stress (*σ_p_*) is applied to calculate the parameters, as described in Equation (3) to Equation (5). According to Equation (3), (4), the constants of *n*_1_ and β could be calculated as shown in Figure 6a–d, which are applied to calculate the constant of α. Then according to Equation (5), the constitutive equation could be derived based on the calculation as shown in Figure 6e,f. Finally, *n*_1_ = 6.75, *β* = 0.034 and *α* = *β/n*_1_ = 0.0050, *n* = 4.636 could be obtained.

By applying linear regression, the activation energy of deformation *Q* = 375.3 kJ/mol for Equation (3), *Q* = 427.0 kJ/mol for Equation (4) and *Q* = 389.1 kJ/mol for Equation (5), as shown in Figure 7. As shown in Figure 7c, the largest large value of R^2^ for the regression indicates a high reliability of the calculated deformation energy according to Equation (5), thus *Q* = 389.1 kJ/mol is believed to be the nominal deformation energy and constitutive relationship between the peak stress, strain rate and deformation temperature could be rewritten in Equation (6) for Fe30Mn11Al1C0.1Nb0.1V.
(6)$$Z=\dot{\varepsilon }exp\left(\frac{389100}{RT\;}\right)=f\left(\sigma \right)=9.52\times 10^{14}{\left[sinh\left(0.0050\sigma_{p}\right)\right]}^{4.636}\;$$

### 4.2. Calculation of the Thermal Processing Map

Processing maps were established in this work by the dynamic material model (DMM), where the mechanical process is considered as a power dissipation system [34]. DMM is suitable for the steel with complex composition, and the hot deformation parameters are related to the evolution mechanism of the internal structure of the steel. The energy dissipation efficiency factor *η* is a dimensionless constant, which reflects the relationship between the energy consumed by the evolution of the internal microstructure and the energy consumed in the ideal linear state. It can be expressed as Equation (7).
(7)$$\eta =\frac{2m}{m+1}\;$$
where *m* is the strain rate sensitivity, which could be calculated by $\sigma =K{\dot{\varepsilon }}^{m}$, *K* is the material constant, and *m* is the sensitive index of strain rate.

In this work, the cubic spline difference method is used to fit the relationship between the function *lnσ* and *ln*$\left(\dot{\varepsilon }\right)$ when the temperature and strain remain unchanged, and Equation (8) is obtained.
(8)$$m=b+2c\ln(\dot{\varepsilon )}$$
where *b* is a constant.

Based on the criterion for microstructural instability at a constant temperature reported by Prasad and Kumar, the condition can be described as Equation (9) [35].
(9)$$\xi \left(\dot{\bar{\varepsilon }}\right)=\frac{\partial ln\left(\frac{m}{m+1}\right)}{\partial ln\dot{\bar{\varepsilon }}}+m<0\;$$
where the dimensionless parameter $\xi \left(\dot{\bar{\varepsilon }}\right)$ describes the locus of flow instability.

According to the above analysis, the processing map generated by the superimposition of power dissipation map and the instability maps of the low-density steel at different strains (*ε* = 0.1, 0.2, 0.4, 0.6, 0.8, 0.9) were obtained, as shown in Figure 8. It can be seen from Figure 8 that the shadowed regions are the instability regions ($\xi \left(\dot{\bar{\varepsilon }}\right)<0$), the efficiencies of power dissipation are represented in percentage by contour numbers in the processing map, which changes with deformation and strain. The efficiency here represents the level of energy dissipation possible under certain deformation conditions. The minimum value of coefficient under different deformations generally appears at the temperature range of 850–950 °C and the strain rate range of 1–10 s^−1^ (gray area B in Figure 8). In this temperature range, the precipitation of carbides at the grain boundary plays a strengthening role and inhibits the softening behavior to a certain extent, which is unfavorable to the hot processing. The maximum efficiency appears at the temperature range of 900–1150 °C and the strain rate range of 0.01–1 s^−1^ (red zone A in Figure 8). It can be seen that the strain variable has certain influence on the dissipation efficiency factor with little significance.

### 4.3. Microstructural and Hardness Evolution during Thermal Processing

The grain refinement microstructure of the Fe30Mn11Al1C0.1V0.1Nb steel developed by hot working contain a certain level density of dislocation, which is caused by work hardening in the relative low temperature of 850-950 °C. The high dislocation density provids driving pressure for the nucleation of grain crystallization. Compared with other Fe-Mn-Al-C austenitic steels, the present samples under discussion contain the dispersed VC and NbC particles. The particles provide pinning pressure retarding the grain boundary motion [36], which is contributed to the grain refinement.

The microhardness and grain size of specimens deformed under different deformation temperatures and strain rates are shown in Figure 9. It can be seen from Figure 9a that the microhardness of the specimen is decreased with increasing deformation temperature. The microhardness of the specimen deformed at the strain rate of 1 s^−1^ under deformation temperatures of 850 °C is 391HV. As the deformation temperatures are increased up to 950 °C, 1050 °C, and 1150 °C at the strain rate of 10 s^−1^, the microhardness of the deformed specimens is decreased to 321HV, 302HV, and 280HV, respectively. There was a similarly change in microhardness at strain rates of 0.01 s^−1^, 0.1 s^−1^ and 1.0 s^−1^ as shown in Figure 2a. The microhardness measured from the deformed specimens indicates that the microhardness is mainly determined by the deformation temperature but not the strain rate, which may be related to the high SFE, low thermal conductivity and the microalloying of Nb and V [37].

Similarly, the average grain size of austenite has a strong dependence of the deformation temperature as shown in Figure 9b. The grain refinement microstructure can be obtained with grain sizes smaller than 3μm at the deformation temperature of 850–950 °C and strain rate of 1-10 s^−1^, which may be resulted from the continuous dynamic recrystallization mechanism during thermal deformation process. The average austenite grain size increases with increasing deformation temperature and decreasing strain rate as shown in Figure 9b. It indicates that the grain refinement microstructure with grain size smaller than 3 μm can be developed in the recrystallized regions of the specimens deformed at 850 °C and 950 °C.

The relationship between the grain size and the microhardness is plotted in Figure 9c. It can be seen that strong dependence of the microhardness and the inverse square-roots of the grain size is described by the classical Hall Petch equation as shown in Equation (10) [38],
(10)$$HV=HV_{0}+k_{HP}d^{-0.5}\;$$
where *HV* is the microhardness, *HV_0_* is the fractional microhardness, *K_HP_* is the Hall Petch slope and *d* is the grain size. Using the measured microhardness and average grain size of specimens deformed under different deformation temperatures and strain rates, the relationship between the microhardness *HV* and the grain size of *d* can be obtained. It can be seen that the strengthening contribution of the grain boundary is manifested by the Hall–Petch hardening slope of 349 HV·μm^0.5^, which is equivalent to about 1047 MPa·μm^0.5^, higher than that of the full recrystallized low-density steel of 719 MPa·μm^0.5^ reported by Massardier-Jourdan et al. [12]. This difference of the Hall Petch slope may be explained by the wide range of twin grain size [12,39] and high dislocation density of present specimens produced by hot compression. It is worthy to be pointed out that this strong dependence of microhardness on the grain size implies that grain refinement high strength low density steel could be developed by the relative low temperature deformation.

## 5. Conclusions

High temperature deformation behaviors and microstructural evolution of a low-density Fe30Mn11Al1C0.1Nb0.1V steel at deformation temperature ranging from 850 °C to 1150 °C and strain rate varying from 0.01 s^−1^ to 10 s^−1^ were investigated with the objective of understanding the deformation’s controlling mechanism and the thermal processing parameters for low density
steel. Conclusions can be drawn as following.


The developed physical-based constitutive equation well predicts the flow behavior of the steel. In this point, the value of activation energy was found to be 389.1 kJ/mol. The constitutive equation for the high temperature deformation of the under discussion low-density steel is:(11)$$\dot{\varepsilon }=4.16384\times 10^{16}\left[sinh{\left(0.0064\sigma \right)}^{6.4499}\right]exp\left(\frac{-385.5299}{8.3145T}\right)\;$$Processing maps were developed at strains of 0.1, 0.2, 0.4, 0.6, 0.8, and 0.9. The domain of safe region is found to be in two parts: one locates at the strain rate range of 1–10 s^−1^ and the temperature range of 900–1150 °C with fully recrystallized microstructure, the other locates at the strain rate range of 0.01-0.1 s^−1^ and the temperature range of 850–950 °C with partial recrystallized microstructure.A grain refinement microstructure can be obtained with the average grain size of 2–4 μm when deformed at relative low temperature 850–950 °C and with relative high deformation strain rates of 1–10 s^−1^, which resulted from the continuous dynamic recrystallization during thermal deformation process. It provides a new grain refinement route to lightweight steel that grain refinement high strength low-density steel can be developed by relative low temperature deformation.A strong Hall Petch relationship between microhardness and grain size of present steel implies a strong strengthening contribution of grain boundary, which is manifested by the high Hall Petch slope of 349 HV·μm^0.5^.


## Figures and Tables

**Figure 1 materials-14-06555-f001:**
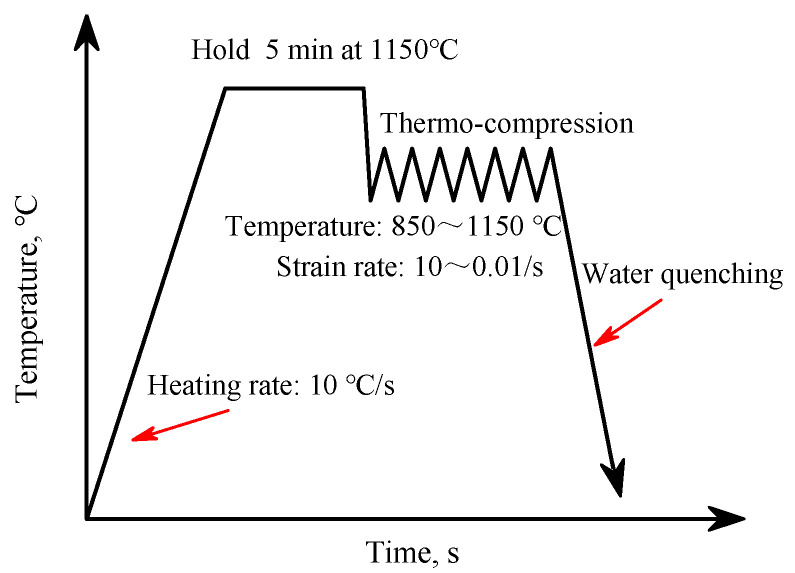
Thermal process diagram of compression tests for the low-density Fe-30Mn-11Al-1C-0.1Nb-0.1V steel.

**Figure 2 materials-14-06555-f002:**
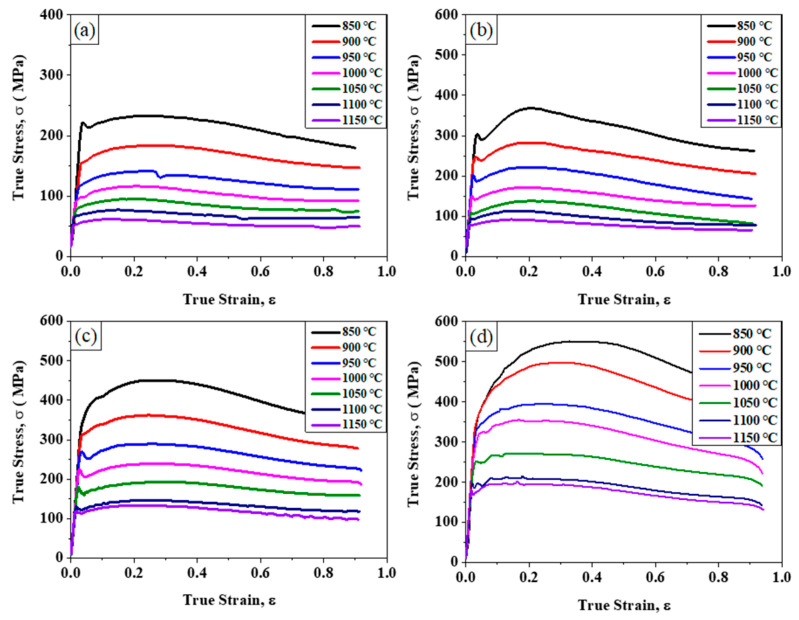
True stress-true strain curves of the experimental steel obtained at different deformation temperatures and under strain rates of (**a**) 0.01 s^−1^, (**b**) 0.1 s^−1^, (**c**) 1 s^−1^ and (**d**) 10 s^−1^.

**Figure 3 materials-14-06555-f003:**
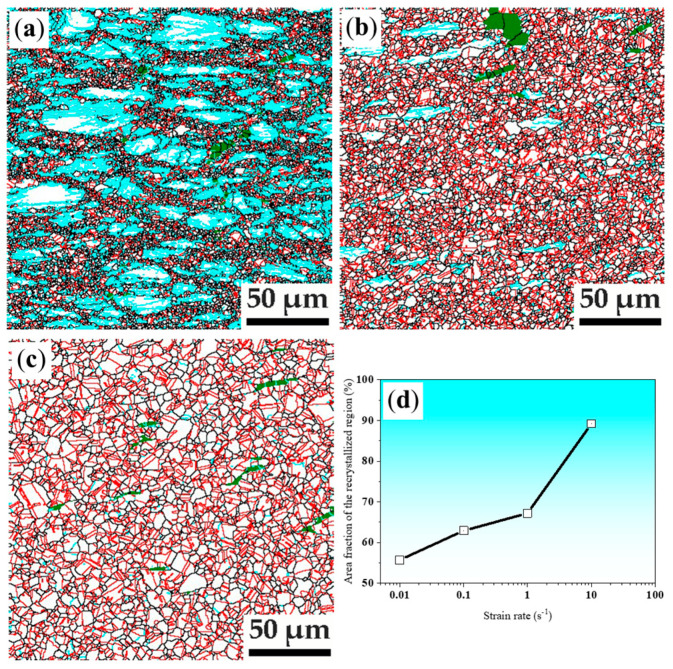
High temperature deformation microstructure revealed by grain boundary maps of specimens compressed at (**a**) 850 °C 1 s^−1^; (**b**) 950 °C, 1 s^−1^; (**c**) 1050 °C, 1 s^−1^. Black lines present high angle grain boundaries (HAGBs: >15°), cyan lines represent low-angle grain boundaries (LAGBs: 2–15°), red lines present twin boundaries (TBs), green color present body centered cubic (BCC) phase; (**d**) Area fraction of recrystallized region versus deformation temperature.

**Figure 4 materials-14-06555-f004:**
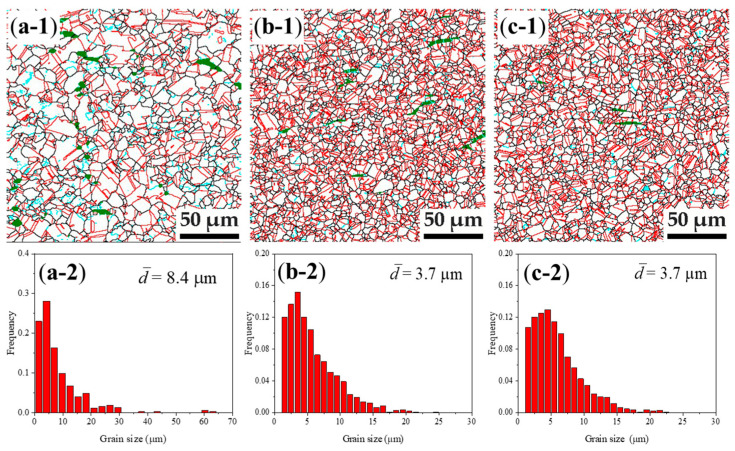
High temperature deformation microstructure: (**a-1**) to (**c-1**) the grain boundary maps, black lines present HAGBs (>15°), cyan lines present LAGBs (2°–15°), red lines present TBs, green color present BCC phase; (**a-2**) to (**c-2**) austenite grain size distribution of specimens compressed at 1050 °C 0.01 s^−1^, 1050 °C 1 s^−1^ and 1050 °C 10 s^−1^, respectively.

**Figure 5 materials-14-06555-f005:**
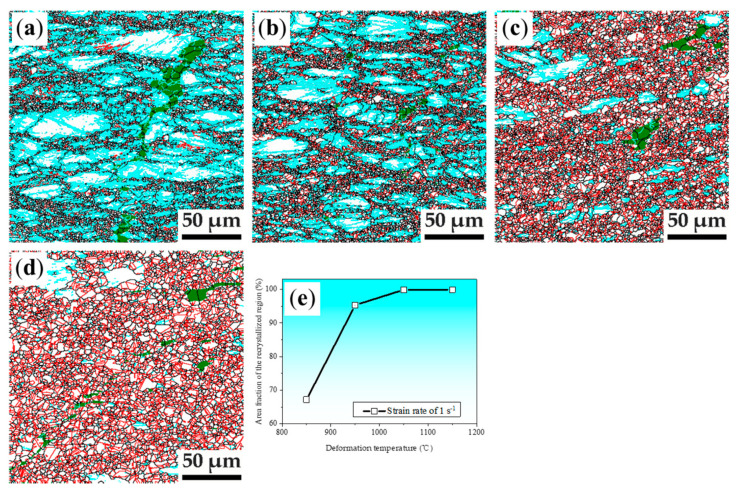
High temperature deformation microstructure revealed by the grain boundary maps of specimens compressed at (**a**) 850 °C, 0.1 s^−1^; (**b**) 850 °C, 1 s^−1^; (**c**) 850 °C, 10 s^−1^; (**d**) 950 °C, 10 s^−1^; black lines present HAGBs (>15°), cyan lines present LAGBs (2–15°), red lines present TBs, green color present BCC phase; (**e**) the area fraction of recrystallized region.

**Figure 6 materials-14-06555-f006:**
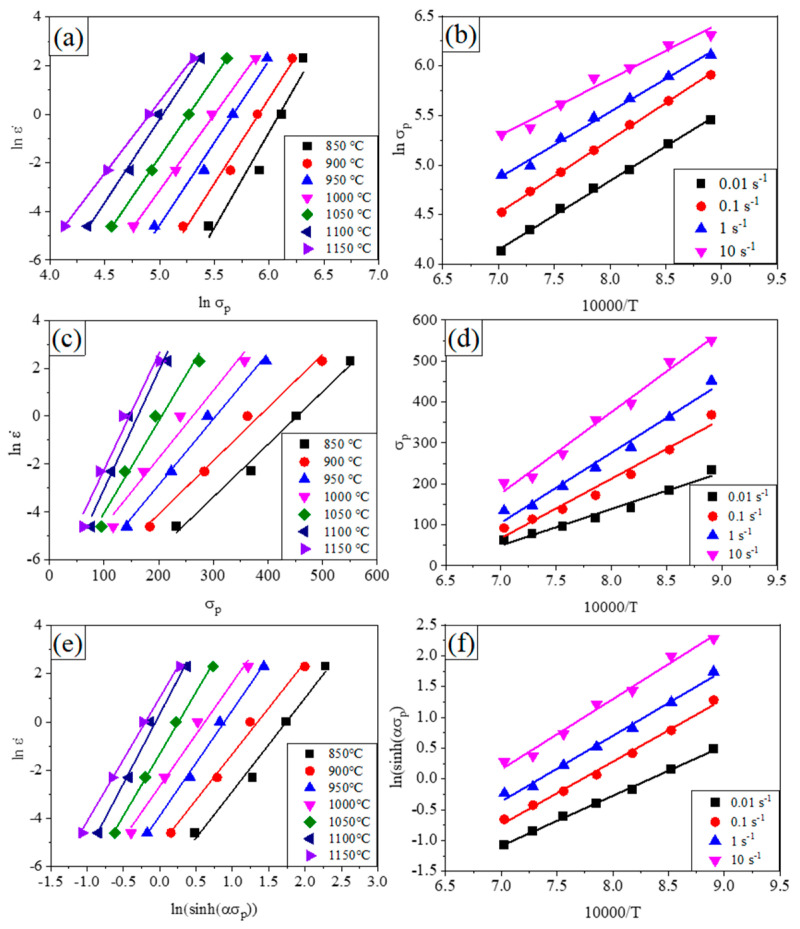
Relationships between the deformation parameters (**a**) *ln(*$\dot{\varepsilon }$*)* and *lnσ_p_*; (**b**) *ln(σ_p_)* and 10,000/T; (**c**) *ln(*$\dot{\varepsilon }$*)* and *σ_p_*; (**d**) *σ_p_* and 10,000/T; (**e**) *ln(*$\dot{\varepsilon }$*)* and *ln(sinh(ασ_p_))*; (**f**) *ln(sinh(ασ_p_))* and 10,000/T.

**Figure 7 materials-14-06555-f007:**
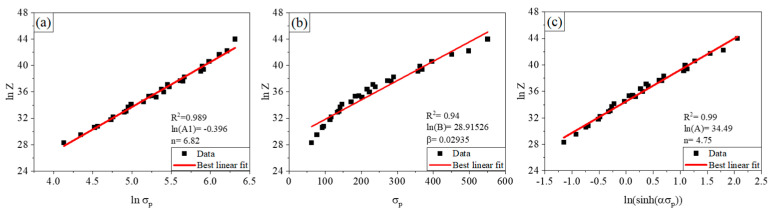
Relationships between the deformation parameters of the low-density steel (**a**) *ln(Z)* and *ln(σ_p_)*; (**b**) *ln(Z)* and *σ_p_*; (**c**) *ln(Z)* and *ln(sinh(ασ_p_))*.

**Figure 8 materials-14-06555-f008:**
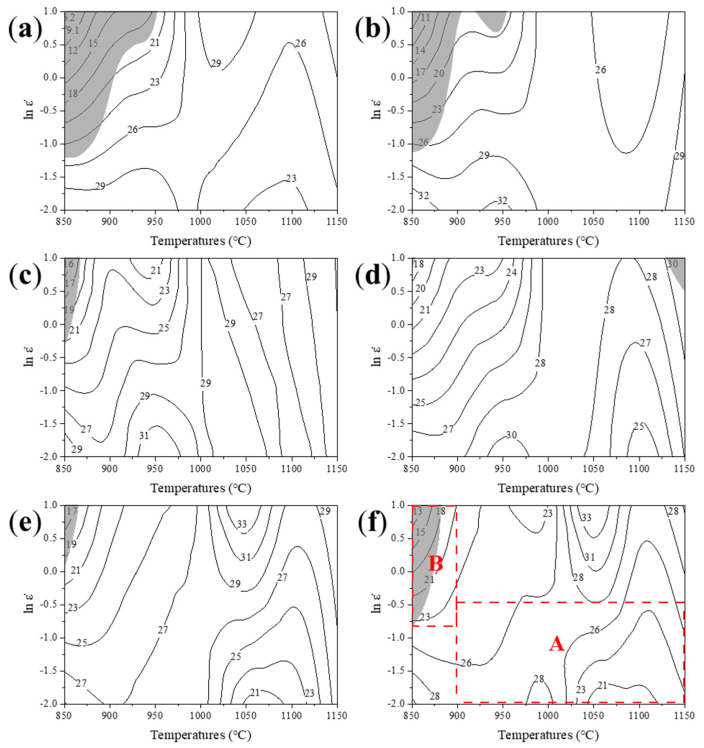
Processing maps for Fe30Mn11Al1C0.1V0.1Nb low density steel at true strains of (**a**) 0.1, (**b**) 0.2, (**c**) 0.4, (**d**) 0.6, (**e**) 0.8 and (**f**) 0.9. The contour numbers represent iso-efficiency of power dissipation expressed in pct.

**Figure 9 materials-14-06555-f009:**
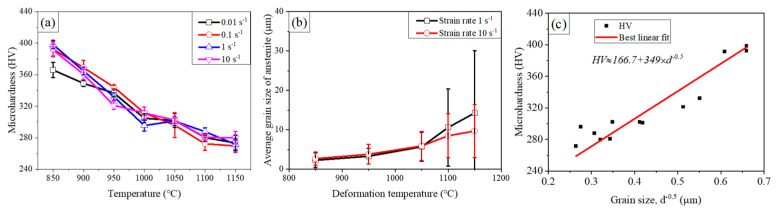
Dependence of microhardness and grain size of Fe-30Mn-11Al-1C-0.1V-0.1Nb low density steel after thermal processing. (**a**) Microhardness after thermal process with different processing conditions, (**b**) Grain size as a function of deformation temperature and (**c**) Relationship between microhardness and grain size.

**Table 1 materials-14-06555-t001:** Chemical composition of Fe-30Mn-11Al-1C-0.1Nb-0.1V steel (wt.%).

Element	C	Mn	Al	Nb	V	Fe
wt.%	1.0	30	11	0.10	0.10	Bal.

## Data Availability

Data sharing is not applicable.

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
