# Peer review of "High Temperature Deformation Behavior and Microstructure Evolution of Low-Density Steel Fe30Mn11Al1C Micro-Alloyed with Nb and V"

_materials, 2021, doi:10.3390/ma14216555_

Round 1
Reviewer 1 Report
The manuscript entitled "High temperature deformation behavior and microstructure evolution of a low-density steel Fe30Mn11Al1C microalloyed with Nb and V" is an interesting experimental study conducted on evaluation of the high temperature deformation behaviour of a low-density steel. However, few issues must be addressed. The paper needs minor revisions before it is processed further, some comments follow:
- There is no clear purpose of the study in the abstract. Please highlight the novelty of the study.
- The introduction section can be improved. In the introduction section, a comprehensive and exhaustive review of the state of the art in the field of the study must be provided. Please refer to previous works, and highlight the experiments and results published previously. In the current form, the introduction section only provides basic/general information about austenitic steel, and the effect of different additions.
- In the section "Discussion" there is no information about new cause-and-effect relationships identified as a result of the study. This does not allow forming an opinion about the scientific novelty and theoretical significance of the research results.
- There is no information about the scientific novelty of the research results in the "Conclusion" section.
- In the "Conclusion" section, there is no information about the prospect of using the research results.
Author Response
Point 1. Abstract:
The selection of the thermal processing parameters is very important to hot rolling and forging process for producing grain refinement in lightweight high manganese and aluminum steels. In this work, the high temperature deformation behaviors of a low-density steel of Fe30Mn11Al1C alloyed with 0.1Nb and 0.1V were studied by isothermal hot compression tests at temperatures of 850 °C - 1150 °C and strain rates between 0.01 s-1 and 10 s-1.
Point 2. Introduction section has been improved as following,
Facing with the requirements of weight-lightening of metallic equipment, low-density steel has been paid much more attention because of its significantly reduced density, high ductility, high toughness and possible wide applications in engineering [1–3]. It has been shown in literatures that the density reduction of the austenitic low-density steel is mainly realized by the lattice expansion and the addition of light elements, such as carbon and aluminium [4–6]. Kalashnikov et al. [7]researched the influence of Al and C content on the tensile strength and U-notch impact toughness of Fe-30Mn-xAl-0.95C austenite steels in 550 °C aged condition. The impact toughness reduced significant with Al content increasing. Acselrad et al.[8] have studied the impact toughness in solutionized and aged condition of Fe-30Mn-10Al-1C-1Si alloy, The materials in the as-solution treated condition show a high charpy V-notch impact energy of 200J with a yield strength of 452Mpa at room temperature, while the impact energy is reduced to almost to zero in aged condition at 550°C for 16h. High Al content make it difficult to obtain both high ductility and high toughness by aging process.
For structural purposes, a good combination of mechanical strength and fracture toughness is desirable. In order to obtain both high ductility and high toughness, the low-density steel has to be solution treated to suppress the occurrence of the κ-carbides, which takes place in the cooling or aging process[2,3,9,10]. At the same time, the solution treatment induces the significant grain growth and therefor decreases both yield and tensile strength according to the classical Hall-Petch law [11,12]. P. Ren et al.[11] produced a series of grain refinement alloys by the cold rolling reduction of 90% and recrystallization annealing, the high yield strength of 945Mpa and 47% elongation were obtained for Fe-30Mn-11Al-1.2C steel with an average grain size of 2.2μm, which is about 350Mpa improvement than the coarsening condition. Thus, it is very important to control the κ-carbide precipitation and austenite grain size by the chemical composition design and the selection of the thermal processing parameters to obtain a very good combination of strength, ductility and toughness.
Point 3. Discussion section:
4.3 has been improved as following:
The grain refinement microstructure of the Fe30Mn11Al1C0.1Nb0.1V steel developed by hot working contain a certain level density of dislocation, which is caused by work hardening in the relative low temperature of 850–950 °C. The high dislocation density provides driving pressure for the nucleation of grain crystallization. Compared with other Fe-Mn-Al-C austenitic steels, the present samples contain the dispersed VC and NbC particles. The particles provide pinning pressure retarding the grain boundary motion[37],which is contributed to the grain refinement.
Point 4. Conclusions point #3 has been improved as following:
- The grain boundary map of specimens compressed at (d)950 °C, 10 s-1 has been added.
Reviewer 2 Report
he manuscript is on topic in the field of lightweight steels. It was obtained new experimental results as well as calculation model. Processing map can be useful and helpful to optimize functional properties of this group of materials. Problematic is well treated. I suggest to add more detail information in introduction to improve state of the art in this field of research. E.g. Nano-mechanical properties of Fe-Mn-Al-C lightweight steels by Alireza Rahnama, or review published by Zambrano: A general perspective of Fe–Mn–Al–C steels, and maybe more.
Author Response
More detail information is added in introduction to improve state of the art in this field of research :
Kalashnikov et al. [7]researched the influence of Al and C content on the tensile strength and U-notch impact toughness of Fe-30Mn-xAl-0.95C austenite steels in 550 °C aged condition. The impact toughness reduced significant with Al content increasing. Acselrad et al.[8] have studied the impact toughness in solutionized and aged condition of Fe-30Mn-10Al-1C-1Si alloy, The materials in the as-solution treated condition show a high charpy V-notch impact energy of 200J with a yield strength of 452Mpa at room temperature, while the impact energy is reduced to almost to zero in aged condition at 550°C for 16h. High Al content make it difficult to obtain both high ductility and high toughness by aging process.
For structural purposes, a good combination of mechanical strength and fracture toughness is desirable. In order to obtain both high ductility and high toughness, the low-density steel has to be solution treated to suppress the occurrence of the κ-carbides, which takes place in the cooling or aging process[2,3,9,10]. At the same time, the solution treatment induces the significant grain growth and therefor decreases both yield and tensile strength according to the classical Hall-Petch law [11,12]. P. Ren et al.[11] produced a series of grain refinement alloys by the cold rolling reduction of 90% and recrystallization annealing, the high yield strength of 945Mpa and 47% elongation were obtained for Fe-30Mn-11Al-1.2C steel with an average grain size of 2.2μm, which is about 350Mpa improvement than the coarsening condition. Thus, it is very important to control the κ-carbide precipitation and austenite grain size by the chemical composition design and the selection of the thermal processing parameters to obtain a very good combination of strength, ductility and toughness.
Reviewer 3 Report
Please take into account the remarks and comments to improve your manuscript.
- Do you have an idea why the amount of recrystallized material is higher with a high strain rate when the deformation temperature is 850°C (Fig. 5). Indeed, with a high strain rate, the duration of the exposition of the steel to the temperature is short which should disfavor the recrystallization process.
- Explain what is the origin of such amount of LAGB is the non-recrystallized parts of the material in Fig 3 and 5
- Figure 9d: it is very strange to fit the data with a straight line. It seems that there are two regimes of hardening versus the grain size. Or it is due to the wide range of grain size at the temperature of 1100 °C and 1150 °C
- Lines 286-289 and Conclusions point #3: it is written: The ultrafine grained microstructure could be obtained with grain size smaller than 3μm when the deformation is carried out in the relative low temperature of 850–950 ℃ but with relative high deformation strain rate of 1–10 s-1.
A histogram giving the distribution of the grain size with the fraction would be more convincing that error bars as it is shown in Figure 9b and 9c.
- Usually ultra-fine grain refers to grain with dimension between 1 and 2 µm.
- Figure 9c: there is no error bars for the test at 1050°C. Can you add to see the net value of the highest grain size?
- Figures 3, 4 and 4 : First check the scale bar where nm is used
- Line 170-171: it is written “The thermal deformation microstructures at both 850 °C and 950 °C examined by EBSD are given in Fig.5.”. But Fig 5 contains only results from tests at 850°C
- Line 72: check the unit for the mass: kg and not Kg
- Figure 8: Refer to zone A and zone B drawn in the figure should be referred in the text
- Some graph titles are written, as follow: deformation strain rate; it should be strain rate
Fig 9b and 9c: change anstenite for austenite in graph title
- Line 143: cyan line present the low
- Line 304: …. reported by Auriane Etienne et al [10]. Mention the first name in the author list of this reference.
Author Response
Thank so much for your valuable advice. I improved my manuscript as follows:
Point 1 and 2:
The grain refinement microstructure of the Fe30Mn11Al1C0.1Nb0.1V steel developed by hot working contain a certain level density of dislocation, which is caused by work hardening in the relative low temperature of 850–950 °C. The high dislocation density provides driving pressure for the nucleation of grain crystallization.
Point 3:
This difference of the Hall-Petch slope may be explained by the fact that the wide range of twin grain size in high temperature[12,40] and dislocation density produced by hot compression.
Point 4:
Conclusions point #3:
- A grain refinement microstructure can be obtained with the average grain size of 2- 4μm when deformed at relative low temperatures of 850–950 ℃ and with relative high deformation strain rates of 1–10 s-1, which may be resulted from the continuous dynamic recrystallization during thermal deformation process. It provides a new grain refinement route to lightweight steels that a high strength low density steel could be developed by the grain refinement through the relative low temperature deformation.
Point 5:
"ultra-fine grain" have been improved as "grain refinement microstructure".
Point 6:
Figure 9c has been delete.
Point 7:
In figure 3 and 4 ,the scale bar has been checked , μm is used now.
Point 8:
The result of test at 950 ℃ has added.
Point 9:
The unit for the mass( kg ) has been checked.
Point 10:
Zone A and B were referred in the text.
Point 11:
Fig 9b and 9c: changed anstenite for austenite in graph title.
Point 12:
It has been modified.
Point 13:
It has been modified.